# Novel Vitamin D3 Hydroxymetabolites Require Involvement of the Vitamin D Receptor or Retinoic Acid-Related Orphan Receptors for Their Antifibrogenic Activities in Human Fibroblasts

**DOI:** 10.3390/cells13030239

**Published:** 2024-01-26

**Authors:** Zorica Janjetovic, Shariq Qayyum, Sivani B. Reddy, Ewa Podgorska, S. Gates Scott, Justyna Szpotan, Alisa A. Mobley, Wei Li, Vijay K. Boda, Senthilkumar Ravichandran, Robert C. Tuckey, Anton M. Jetten, Andrzej T. Slominski

**Affiliations:** 1Department of Dermatology, University of Alabama at Birmingham, Birmingham, AL 35294, USA; zjanjetovic@uabmc.edu (Z.J.); sqhusain@gmail.com (S.Q.); sbreddy@gmail.com (S.B.R.); epodgorska@uabmc.edu (E.P.); scotts@acom.edu (S.G.S.); j.szpotan@wp.pl (J.S.); alisamo@uab.edu (A.A.M.); sravichandran@uabmc.edu (S.R.); 2Brigham’s Women’s Hospital, Harvard University, Boston, MA 02115, USA; 3College of Pharmacy, The University of Tennessee Health Science Center, Memphis, TN 38163, USA; wli@uthsc.edu (W.L.); vb@uthsc.edu (V.K.B.); 4School of Molecular Science, The University of Western Australia, Perth 6009, Australia; robert.tuckey@uwa.edu.au; 5Cell Biology Section, NIEHS, National Institutes of Health, Research Triangle Park, NC 27709, USA; jetten@niehs.nih.gov; 6Cancer Chemoprevention Program, Comprehensive Cancer Center, University of Alabama at Birmingham, Birmingham, AL 35294, USA; 7VA Medical Center, Birmingham, AL 35294, USA

**Keywords:** vitamin D3, metabolites, fibroblasts, vitamin D receptor, retinoic acid-related orphan receptors

## Abstract

We investigated multiple signaling pathways activated by CYP11A1-derived vitamin D3 hydroxymetabolites in human skin fibroblasts by assessing the actions of these molecules on their cognate receptors and by investigating the role of CYP27B1 in their biological activities. The actions of 20(OH)D3, 20,23(OH)_2_D3, 1,20(OH)_2_D3 and 1,20,23(OH)_3_D3 were compared to those of classical 1,25(OH)_2_D3. This was undertaken using wild type (WT) fibroblasts, as well as cells with *VDR*, *RORs*, or CYP27B1 genes knocked down with siRNA. Vitamin D3 hydroxymetabolites had an inhibitory effect on the proliferation of WT cells, but this effect was abrogated in cells with silenced *VDR* or *RORs.* The collagen expression by WT cells was reduced upon secosteroid treatment. This effect was reversed in cells where VDR or RORs were knocked down where the inhibition of collagen production and the expression of anti-fibrotic genes in response to the hydroxymetabolites was abrogated, along with ablation of their anti-inflammatory action. The knockdown of *CYP27B*1 did not change the effect of either 20(OH)D3 or 20,23(OH)_2_D3, indicating that their actions are independent of 1α-hydroxylation. In conclusion, the expression of the VDR and/or RORα/γ receptors in fibroblasts is necessary for the inhibition of both the proliferation and fibrogenic activity of hydroxymetabolites of vitamin D3, while CYP27B1 is not required.

## 1. Introduction

Vitamin D3, a pro-hormone, is photosynthesized in the skin after exposure to ultraviolet B wavelengths (UVB) of solar radiation [1,2,3,4,5,6]. It can be activated in the classical pathway trough hydroxylation at C25 and C1α, the latter mediated by CYP27B1, to produce biologically active 1,25-dihydroxyvitamin D3 (1,25(OH)_2_D3) [2,6,7,8,9]. An alternative pathway of vitamin D activation involves its hydroxylation by CYP11A1 to produce 20-hydroxyvitamin D3 (20(OH)D3) as the major product. This can be further hydroxylated to 20,23(OH)_2_D3 by CYP11A1, or to other di- and tri-hydroxy metabolites by other CYPs [10,11,12,13]. Skin fibroblasts express several of these CYPs that can metabolize vitamin D derivatives, including CYP11A1 which is involved in this alternative pathway of vitamin D activation [12,14,15], as well as CYP27B1 which is responsible for 1α-hydroxylation [16]. The novel vitamin D3 hydroxyderivatives produced by CYP11A1 inhibit skin cell proliferation and inflammation and have anticancer activities in the skin, similar to classical 1,25(OH)_2_D3 [17,18,19,20], as well as acting as photoprotective agents [21]. These secosteroids exhibit anti-fibrotic activity and downregulate the synthesis of collagen in fibroblast cells [22,23]; yet, unlike 1,25(OH)_2_D3, they are not calcemic and are non-toxic [17,23,24,25]. Vitamin D deficiency has many health consequences [3,6,26,27], including involvement in the pathogenesis of fibrosis, particularly common in chronic liver disease [28] and in systemic sclerosis [29]. 

The vitamin D receptor (VDR) is a nuclear receptor with the primary role of regulating calcium homeostasis and is activated by 1,25(OH)_2_D3, as well as other vitamin D hydroxyderivatives [3,30,31,32,33,34,35,36]. The nuclear retinoic acid-related orphan receptors (RORs) (RORα and RORγ) are expressed in many cells, including skin and various immune cells, and are involved in the pathogenesis of inflammatory diseases, as well as many malignant processes [37,38,39]. They are the targets for vitamin D derivatives [33,37,40], which act as inverse agonists and can also modulate collagen production [41,42,43,44]. 

Fibrosis is a pathological process characterized by abnormal synthesis and the subsequent accumulation and deposition of collagen, leading to an impaired function of the affected organ, such as the skin or lungs [45]. This chronic disease is known as scleroderma or systemic sclerosis [46] and is characterized by autoimmunity, high collagen formation and fibroblast activation. Fibroblasts are highly active cells found in abundance in connective tissue, where they synthesize pro-collagen molecules and are involved in the regulation of the expression of matrix metalloproteinases (MMPs) and inhibitors of metalloproteinases (TIMPs) [47]. In normal tissue, fibroblasts are involved in wound healing and repair; in patients with sclerotic diseases they are excessively activated to produce type I and type III collagen, matrix metalloproteinases (MMPs), profibrotic cytokines and transforming growth factor-β1 (TGF-β1), thus contributing to fibrosis [46,47]. Primary dermal fibroblasts can produce biologically active vitamin D3 hydroxymetabolites, including 20(OH)D3, 20,23(OH)_2_D3 and 1,25(OH)_2_D3 [14,16]. Vitamin D3 metabolites, especially hormonally active 1,25(OH)_2_D3, have been used for regulating gene expression through interaction with the VDR, for disease treatment, including fibrosis [16,41,44,48,49,50]. A plethora of studies have demonstrated the inhibitory effect of 1,25(OH)_2_D3 in fibrotic skin diseases [28,44,48,49,51,52]. In addition, several D3 metabolites have been tested on fibroblasts, including 20(OH)D3 and 20,23(OH)_2_D3, the major products of CYP11A1-mediated vitamin D3 metabolism [17,23,53,54]. Both metabolites suppress fibrogenesis and cell proliferation, as tested on murine skin and human skin cells [17,23,55]. The exact mechanism underlying the development of systemic sclerosis is still under investigation. There are many pathways proposed to be involved in fibrotic tissue formation, resulting in different treatment modalities, including the use of vitamin D derivatives to target the VDR [56]. Unfortunately, none of the available treatment modalities are fully effective. 

Our recent study on RORγ-deficient murine skin cells revealed that the antifibrogenic activities of the novel CYP11A1-derived secosteroids are dependent on that receptor [55]. In the current study, we have further investigated the roles of the VDR and RORs in fibrosis, using human dermal fibroblasts with silenced *VDR*, *RORA* or *RORC* genes. We hypothesized that the action of the novel vitamin D derivatives, 20(OH)D3, 20,23(OH)_2_D3, 1,20(OH)_2_D3 and 1,20,23(OH)_3_D3, on human fibroblasts would depend on the VDR and/or RORα/γ for their inhibition of pro-fibrotic activities. We have also tested whether CYP27B1, the enzyme catalyzing the hydroxylation of vitamin D derivatives at C1α, is necessary for the anti-fibrotic activity of 20(OH)D3 and 20,23(OH)_2_D3 in human fibroblasts.

## 2. Materials and Methods

### 2.1. Cell Culture and Treatment

Skin tissues from the neonatal foreskin of either black or white donors were used to freshly isolate fibroblasts (HDFn), as previously described [17]. Cells were cultured in complete Dulbecco’s modified eagle medium (DMEM) media (Cellgro Technologies, Lincoln, NE, USA) containing 10% serum (Sigma Chemical Co., St Louis, MO, USA), and 1% antibiotic (Cellgro Technologies, Lincoln, NE, USA). During treatment, cells were grown in the presence of 10% charcoal-stripped serum (Sigma Chemical Co., St Louis, MO, USA), instead of complete serum. Cells isolated from different skin tissues within the same race were combined for greater biological variability. Cells were passaged three–eight times to achieve an adequate amount with high confluence. Cells were serum-deprived overnight for synchronization and subsequently treated with the vitamin D3 hydroxyderivatives 20(OH)D3, 20,23(OH)_2_D3, 1,20(OH)_2_D3 or 1,20,23(OH)_3_D3, added from ethanol stock solutions. Recombinant human TGF-β1 (R&D Systems, Minneapolis, MN, USA) (final concentration 5 ng/mL) was added two hours post treatment for the measurement of collagen expression and fibrosis markers, as previously described. 

The CYP11A1-derived vitamin D3 hydroxyderivatives, including 20(OH)D3, 20,23(OH)_2_D3, 1,20(OH)_2_D3 and 1,20,23(OH)_3_D3, were synthesized and purified as described previously [57,58,59]. In addition, commercially available 1,25(OH)_2_D3 (Sigma Chemical Co., St Louis, MO, USA) and ethanol were used as positive or negative controls, respectively. 

For gene silencing, HDFn were plated onto 150 mm Petri dishes (Corning Cellgro, Manassas, VA, USA) and cultured in full media until they reached 50% confluence. Lafayette, Next, cells were transfected with 15 nM of Accell SMARTpool siRNAs (Dharmacon Inc., CO, USA), targeting human *RORC* (RORγ), *RORA* (RORα), *VDR*, or *CYP27B1* in the siRNA delivery medium. *GAPDH* and non-target control RNA were used as a positive and negative control, respectively. After 96 h of incubation, cells were passaged and plated onto different plates, depending on the type of experiment. Both Western blotting and RT-qPCR were used to evaluate the efficiency of the knockdown of gene expression in cells passaged three times post-siRNA transfection.

### 2.2. Western Blot Analyses Found That the Vitamin D Receptor Is Required for the Action of Vitamin D Hydroxyderivatives on Fibroblasts

Western blot analyses were carried out 96 h after siRNA transfection by using 20 μg of proteins from each sample, isolated from transfected cells that had been lysed in RIPA lysis buffer. Proteins were separated using a Mini-PROTEAN^®^TGX™gel (BioRad, Hercules, CA, USA). The blots were first probed with the antibody against the protein of interest. Next, blots were stripped and re-probed with anti-beta-actin antibody [20,21]. The primary antibodies used were mouse monoclonal antibody against VDR (sc-13133, Santa Cruz Biotechnology, Dallas, TX, USA; 1:1000 dilution), goat polyclonal antibody against CYP27B1 (sc-49642, Santa Cruz Biotechnology, Dallas, TX, USA; 1:200), rat monoclonal antibody against ROR gamma (14-6988-82, Invitrogen, Carlsbad, CA, USA, 1:1000), ROR alpha mouse monoclonal antibody, clone OTI5A6 (TA803241S OriGene, Rockville, MD, USA; 1:2000) or mouse monoclonal anti-betaactin peroxidase antibody (A3854, Sigma, Burlington, MA 1:5000). Blots were incubated overnight at 4 °C with the appropriate antibodies. After incubation, membranes were washed three times for 10 min, then incubated for 2 h at room temperature with the HRP-conjugated secondary antibodies, either anti-mouse secondary antibody (sc-516102), mouse anti-goat secondary antibody (sc-2354, at 1:3000 dilution), or anti-rat antibody (sc-2750, at 1:3000) (all from Santa Cruz Biotechnology, Dallas, TX, USA). Immuno-reactivity was detected using a SuperSignal West Pico ECL (BioRad, Hercules, CA, USA). The protein bands of interest were identified according to their molecular weights (kDa) published in the manufacturers’ datasheets, determined relative to the precision plus protein™ kaleidoscope™ standards (BioRad, Hercules, CA, USA). Band intensities, measured by ImageJ, were normalized relative to the loading control and to the band intensities seen in control cells (positive or negative siRNA control), and are presented as % of control. RT-qPCR was performed to confirm the knockdown of the target gene expression for every experiment (see below).

### 2.3. Cell Proliferation Mediators Retinoic Acid-Related Orphan Receptor-α and Retinoic Acid-Related Orphan Receptor-γ Are Required for the Effects of Vitamin D Hydroxyderivatives on Fibroblasts

For proliferation studies, fibroblasts were plated onto 96-well plates and treated with the listed vitamin D hydroxymetabolites as previously described, at concentrations of 1, 10 or 100 nM, in six replicas. Cells were incubated for 20 or 44 h, after which 20 μL of 3-(4,5-dimethylthiazol-2-yl)-5-(3-carboxymethoxyphenyl)-2-(4-sulfophenyl)-2H-tetrazolium, inner salt (MTS) reagent/well (Promega, Madison, WI, USA) solution was added to the media, and the plates were incubated for an additional 3 h. The absorbance was recorded at 490 nm.

### 2.4. Ribonucleic Acid Isolation and Quantitative Reverse Transcription Polymerase Chain Reaction

Fibroblasts were cultured on 60 mm diameter Petri dishes. Once the cells reached 95% confluence, they were treated with secosteroids (100 nM final concentration) diluted in media containing stripped serum, one plate per treatment, per experiment. Cells were harvested 24 h later for RNA isolation and supernatant was collected for collagen assays. Total RNA was isolated using an Absolute RNA miniprep kit (Agilent Technologies, Santa Clara, CA, USA), and quantified using a NanoDrop-2000 (Biotek, NJ, USA) spectrophotometer. cDNA was synthesized using a cDNA synthesis kit (Transcriptor First Strand cDNA Synthesis Kit, Roche, Indianapolis, IN, USA). PCR was performed in triplicates with Luminaris HiGreen low ROX PCR reagent (ThermoFisher, Waltham, MA, USA), as previously described. Samples were normalized to glyceraldehyde-3-phosphate dehydrogenase (GAPDH) or Cyclophilin B mRNA expression levels. Data for fibroblasts treated with vitamin D3 hydroxyderivatives were calculated as ΔΔCT normalized to an endogenous reference and analyzed using GraphPad Prism 9 statistical software and the *t*-test. Data are presented as the fold-change in gene expression in comparison to the vehicle (EtOH) in graphical form. Student’s *t*-test was used for statistical analysis (* *p* < 0.5; ** *p* < 0.1). Table 1 lists the primer sequences used for PCR.

### 2.5. Collagen Assay

The supernatant, which was derived from fibroblasts covered with 3 ml of media, was collected 24 h post-treatment to measure total collagen production. The collagen isolation and measurement of its concentration were performed using a Sircol collagen assay kit (Biocolor Ltd., Carrickfergus, UK). The absorbances of the samples were recorded at 555 nm and the values were compared to a collagen standard curve, to calculate the concentration of total collagen.

## 3. Results

### 3.1. Silencing of the Vitamin D Receptor, Retinoic Acid-Related Orphan Receptors, or CYP27B1 

To decipher the mechanism of the anti-fibrotic activity of 20(OH)D3 and 20,23(OH)_2_D3, along with their CYP27B1-derived metabolites 1,20(OH)_2_D3 and 1,20,23(OH)_3_D3, we investigated the involvement of VDR, RORα and RORγ receptors, as well as CYP27B1 enzyme-dependent pathways. These receptors appear to be important for the action of many vitamin D3 metabolites [19,23,37,55,58,66]. Using human dermal fibroblasts (HDFn), we optimized the Accell protocol for silencing the receptors using siRNA. Initially, HDFn were transfected with a range of concentrations of SMARTpool siRNAs, as recommended by the manufacturer. The knockdown efficiency was evaluated by Western blotting and RT-qPCR. The best silencing (>75%) of the expression of all the proteins under study, compared to non-targeted siRNA controls, was observed with 15 nM of each siRNA (Figure 1). This concentration was therefore used in subsequent experiments for all knockdowns. All experiments were performed on fibroblasts isolated from white or black donors. Both showed similar effects, suggesting that skin color does not play a role in the effects of vitamin D on fibroblasts.

### 3.2. The Vitamin D Receptor Is Required for the Action of Vitamin D Hydroxyderivatives on Fibroblasts

Initially, we examined whether the ability of vitamin D3 metabolites to inhibit fibroblast proliferation and collagen formation was mediated by the VDR, as previously reported for the action of 20(OH)D3 on immortalized keratinocytes [67]. Both wild type human dermal fibroblasts and fibroblasts in which the VDR was silenced (referred to as si-VDR) were prepared for proliferation assay and treated with 20(OH)D3, 20,23(OH)_2_D3, 1,20(OH)_2_D3, 1,20,23(OH)_3_D3 or 1,25(OH)_2_D3 (positive control). Ethanol was used to dissolve the vitamin D3 metabolites and was therefore used as a negative control (vehicle at 0.1%). Cells were exposed to 1, 10 or 100 nM of the vitamin D hydroxyderivatives for 24 or 48 h. Since some vitamin D3 metabolites can inhibit TGF-β1-induced type I collagen production [68], we also added TGF-β1, a potent fibrogenesis inducer, to the cells, 2 h after the addition of vitamin D3 hydroxyderivatives. As shown in Figure 2A, all the secosteroids tested inhibited cell proliferation in control fibroblasts but not in si-VDR fibroblasts. Significant differences (*p* < 0.05 or less) were seen when comparing the level of proliferation between control and si-VDR cells at secosteroid concentrations of 10 and 100 nM. For the si-VDR cells, there were no significant differences between the secosteroid-treated and the vehicle-treated cells. These data indicate that the action of vitamin D metabolites on the proliferation of human fibroblasts is dependent on a functional VDR.

The data in Figure 2B show that all vitamin D3 hydroxyderivatives tested, 20(OH)D3, 20,23(OH)_2_D3, 1,20(OH)_2_D3 and 1,20,23(OH)_3_D3, decreased the amount of total collagen in the supernatant derived from WT cells at 24 h to a similar extent to 1,25(OH)_2_D3 (*p* < 0.01), which has known antifibrotic activity. For the si-VDR fibroblasts, the amount of collagen detected in supernatants for all secosteroid treatments was the same as in vehicle-treated control cells (Figure 2B). This indicates that the VDR is necessary for the anti-fibrotic action of vitamin D hydroxyderivatives.

We confirmed these findings by measuring expression genes encoding collagen using RT-qPCR. The expression of collagen genes *COL1A1* and *COL1A2* was decreased in WT fibroblasts with all secosteroids tested except 20(OH)D3 on COL1A2 (Figure 2C). Generally, there was little change or an increase in *COL1A1* and *COL1A2* gene expression in si-VDR cells, compared to vehicle-treated controls, for the secosteroids tested. The expression of the fibronectin gene (*FN1*) decreased after treatment with all the vitamin D hydroxymetabolites, and this effect was abrogated in cells lacking VDR, other than for 20(OH)D3, where the levels remained lower than in vehicle-treated cells, regardless of VDR knockdown. Similar results were seen for the expression of the smooth muscle actin gene (*ACTA1*), which decreased in control cells after secosteroid treatment and also in si-VDR cells after treatment with 20(OH)D3 or 20,23(OH)_2_D3, but was similar to vehicle-treated control cells for the si-VDR cells treated with 1,20(OH)_2_D3 or 1,20,23(OH)_3_D3 (Figure 2C). 

Selected anti-inflammatory genes, *IL-6*, *IL-8* and *TGFB1*, also showed some changes in expression in response to secosteroids in control and si-VDR cells (Figure 2C). The level of *IL-6* expression decreased in control cells in response to all the secosteroids tested, whereas it increased with all treatments in the si-VDR cells relative to the vehicle control. The results indicate that the anti-inflammatory effect on *IL-6* expression of the vitamin D derivatives requires the VDR. On the other hand, the VDR appears to be less important for the action of vitamin D3 hydroxymetabolites on levels of *IL-8* expression, where the greatest effect (a decrease) was generally seen in the si-VDR cells. Only small and variable effects were seen for the secosteroids on the expression of the transforming growth factor beta 1 gene (*TGFB1*) in the control cells, with a variable response also seen in si-VDR cells compared to the vehicle control. Changes in gene expression in response to the secosteroids in si-VDR cells, particularly for *IL6* and *IL8* expression, suggest that other receptors besides the VDR may be involved in the anti-inflammatory action of the vitamin D3 hydroxymetabolites.

### 3.3. Retinoic Acid-Related Orphan Receptor-α and Retinoic Acid-Related Orphan Receptor-γ Are Required for Phenotypic Effects of Vitamin D Hydroxyderivatives on Fibroblasts

RORs are expressed in human skin, and CYP11A1-derived vitamin D3 hydroxymetabolites act as inverse agonists on them [37]. We previously tested the involvement of RORγ in the anti-fibrotic action of 20(OH)D3 in mice [55] and showed that the anti-proliferative and anti-fibrotic activities of the vitamin D3 hydroxymetabolites are RORγ-dependent. In this study, we expanded our analysis to human fibroblasts in which either RORα or RORγ expression was knocked down by siRNA (referred to as si-RORα and si-RORγ) (Figure 1). Dermal fibroblasts were treated with 20(OH)D3, 20,23(OH)_2_D3, 1,20(OH)_2_D3 or 1,20,23(OH)_3_D3, at concentrations ranging from 1 to 100 nM, for 24 or 48 h. The ethanol vehicle was used as a negative control and 1,25(OH)_2_D3 as a positive control. There was a strong, concentration-dependent inhibitory effect of all the secosteroids on the proliferation of WT human fibroblasts (*p* < 0.05 or less) (Figure 3A). However, the knockdown of either RORα or RORγ (si) removed the inhibitory effects of the secosteroids on fibroblast proliferation (Figure 3A). Thus, both RORγ and RORα appear to be involved in the action of the CYP11A1-derived vitamin D derivatives to regulate the proliferation of human fibroblasts, in agreement with a previous study that showed that these secosteroids act through RORγ on murine fibroblasts [55].

To investigate the role of RORα/γ receptors in mediating the effect of CYP11A1-derived vitamin D3 hydroxymetabolites on collagen synthesis, we examined wild type, si-RORα and si-RORγ human fibroblasts. For the control cells, a significant decrease in collagen content (*p* < 0.001) compared to the vehicle control was observed for all secosteroids. In contrast, no effect was seen for the secosteroids in si-RORα and si-RORγ cells, while an increase in collagen occurred only with 20(OH)D3 (Figure 3B). Thus, both RORα and RORγ are required for the anti-fibrotic effects of the vitamin D3 metabolites. To obtain further support for the role of RORα/γ receptors in human fibroblasts, we examined the expression of several genes involved in fibrosis and inflammation after treatment with vitamin D3 hydroxymetabolites. The graphs in Figure 3C show that the expression of genes involved in fibrosis (*COL1A1, COL1A2, COL3A1*, *FN1, THBS1, PDGFA* and *ACTA1*)*,* and inflammation (*IL6, IL8, IL33* and *TGFB1*) were generally reduced by the secosteroids in WT cells, compared to the vehicle control. As expected, *MMP1* expression was two- to five-fold higher in control cells in response to the secosteroids. In contrast, in si-RORα cells, the secosteroids generally caused only minimal change or a moderate increase in the expression of genes involved in fibrosis but caused a large increase in the expression of some genes involved in inflammation. There was some variability in the effects between the different secosteroids, with one example of this being 1,20,23(OH)_3_D3, which did not always show changes observed with the other secosteroids. The results for the si-RORγ cells were similar to those for si-RORα, with a marked increase in the expression of *FN1* by all secosteroids and increased expression of *THBS1*, *IL6, IL8* and *IL33*. Lastly, while *MMP1* was highly expressed in control cells after treatment with vitamin D3 hydroxymetabolites, this effect was abrogated in si-RORα and si-RORγ cells, with some secosteroids even causing a decrease in *MMP1* expression (Figure 3C). Overall, these results show that RORs are involved in the regulation of the expression, by secosteroids, of genes associated with both fibrosis and inflammation.

### 3.4. 1α-Hydroxylation Is Not Necessary for the Action of 20(OH)D3 or 20,23(OH)_2_D3 on Fibroblasts

The results above demonstrate that 20(OH)D3, 20,23(OH)_2_D3, 1,20(OH)_2_D3 and 1,20,23(OH)_3_D3 exhibit antiproliferative and antifibrotic activities in fibroblasts (Figure 4). 20(OH)D3 and 20,23(OH)_2_D3 are the major products of the CYP11A1-induced metabolism of vitamin D3, with some being converted to 17,20,23(OH)_3_D3 (Figure 4A) [11,13,69]. CYP27B1 plays a key role in activating 25(OH)D3, the major circulating form of this vitamin, by hydroxylating it at the C1α-position to produce 1,25(OH)_2_D3 [3,27,69,70]. It is not known whether a similar 1α-hydroxylation of 20(OH)D3 and 20,23(OH)_2_D3 is required for their action on fibroblasts (Figure 4A). CYP27B1 has been reported to catalyze this reaction in vitro in studies using purified enzymes [71] and in placental tissue [13] and skin cells [23]. To determine whether the activities of 20(OH)D3 and 20,23(OH)_2_D3 are regulated by hydroxylation at C1α, we used siRNA to silence the *CYP27B1* gene in fibroblasts (referred to as si-CYP27B1), with negligible CYP27B1 being expressed in these cells as judged from the Western blot (Figure 1). 

As expected, a strong, concentration-dependent inhibitory effect on proliferation was observed in control cells treated with either 20(OH)D3 or 20,23(OH)_2_D3, when compared to the vehicle control (Figure 4B). This effect was also seen in si-CYP27B1 fibroblasts, and there were no significant differences in the degree of inhibition between control and si-CYP27B1 cells at any of the concentrations tested. Both secosteroids reduced the amount of soluble collagen present in the cell supernatants, regardless of whether *CYP27B1* was expressed or not (*p* < 0.05) (Figure 4C). These results indicate that the expression of CYP27B1 is not necessary for the action of 20(OH)D3 or 20,23(OH)_2_D3 on proliferation or collagen synthesis in human fibroblasts and that these secosteroids do not require 1α-hydroxylation for their biological activity. 

Finally, we investigated the role of CYP27B1 in the action of 20(OH)D3 and 20,23(OH)_2_D3 on the expression of fibrosis-related genes. Figure 4D shows that the expression of *COL1A1*, *COL3A1*, *TGFB1* and *THBS1* was reduced by the two secosteroids in both control and silenced cells, indicating no requirement for CYP27B1, with COL1A2 only being reduced in control cells.

## 4. Discussion

Fibrosis is a physiological process involved in wound healing; however, hyper-proliferation of fibroblasts and overproduction of collagen leads to tissue scarring [47,56]. Treatment with vitamin D can lessen fibrosis [44,46,48,50,51]. We have discovered several CYP11A1-derived vitamin D3 hydroxymetabolites, some of which are naturally found in human serum [72,73], in human organs or cells [13,14] and in bees’ honey [74], with metabolites lacking a hydroxyl group on C1α being non-calcemic [24,25]. We previously reported that these vitamin D3 hydroxymetabolites, similarly to 1,25(OH)_2_D3 [52], have the ability to reduce fibrosis in human cells [22,23] or murine skin in vivo [17,53]. In mice, this effect required the expression of the RORγ receptor [55]. The CYP11A1-derived vitamin D3 hydroxymetabolites act as inverse agonists on RORs [37]. In addition to RORs [37], we also reported that the VDR is a target receptor for the CYP11A1-derived vitamin D derivatives [19,34,58], resulting in the regulation of various biological processes, including anti-inflammatory effects. CYP11A-derived 20(OH)3 and 20,23(OH)_2_D3 can undergo further metabolism by CYP11A1 and by other vitamin D3-metabolizing CYPs [75]. An important example is CYP27B1, which acts as a 1α-hydroxylase, thus producing 1,20(OH)_2_D3 and 1,20,23(OH)_3_D3, respectively (Figure 4A), which also are biologically active in skin cells [13,57,71]. To unravel the underlying mechanism of action of CYP11A1-derived vitamin D3 hydroxymetabolites in fibrosis, we investigated the involvement of the VDR- and ROR-dependent pathways, and tested whether CYP27B1 is necessary to activate the two secosteroids lacking a C1α-hydroxyl group, namely 20(OH)D3 and 20,23(OH)_2_D3. To achieve this goal, we silenced the expression of the VDR, RORα, RORγ or CYP27B1 in human fibroblasts using siRNA, with a high degree of knockdown being achieved. We did not observe any significant difference in the proliferation of the non-treated cells, in either the presence or absence of the expression of the specified genes.

The control (WT) fibroblasts treated with 20(OH)D3, 20,23(OH)_2_D3, 1,20(OH)_2_D3 or 1,20,23(OH)_3_D3 exhibited decreased proliferation, but this effect was abrogated in si-VDR cells. Because vitamin D3 is known to decrease fibrosis by reducing collagen expression [44], we tested the effects of the secosteroids under study on collagen production by si-VDR fibroblasts, compared to control cells. The expression of collagen was reduced in control cells but not in si-VDR fibroblasts, with similar results being seen for 1,25(OH)_3_D3. We further explored the role of VDR in fibrosis and inflammation by analyzing the expression of several fibrosis-related genes in control and si-VDR fibroblasts treated with the CYP11A1-derived secosteroids, by RT-qPCR. Collagens are most commonly found in connective tissue being produced by fibroblasts [76], and the downregulation of their expression (e.g., *COL1A1* and *COL1A2*) was seen in control fibroblasts treated with the novel vitamin D3 hydroxymetabolites, whereas their expression was relatively unchanged or increased in the si-VDR fibroblasts, depending on the individual secosteroid. Furthermore, the decrease in the expression of several other genes involved in fibrosis, such as *FN1*, encoding a growth factor for fibroblasts [77], and *ACTA1*, encoding an important factor in the regulation of fibroblasts function, appeared to require the VDR. The expression of *TGFB1,* an important marker of tissue fibrosis, as well as inflammation [78], was not markedly affected by the secosteroids and no clear requirement for the VDR was seen, with variability noted between the different secosteroids, likely related to the contradictory role of *TGFB1* (pro- or anti-inflammatory) [79]. The involvement of the VDR in the anti-inflammatory effects of vitamin D3 is shown by the decrease in *IL-6* gene expression in control fibroblasts, but not in si-VDR cells. In fact, knockdown of the VDR led to an increase in *IL-6* expression after treatment, suggesting the involvement of other nuclear receptors on which CYP11A1-derived secosteroids can act with selectivity, defined by the position of the hydroxyl group either on the side chain or at C1α. The expression of *IL-8* in both control and si-VDR cells was generally reduced, also suggestive of the involvement of other nuclear receptors. In summary, our study suggests that VDR plays an important role in the regulation of proliferation and collagen expression in human fibroblasts; however, this role is not absolute. 

Previous data revealed that RORα and γ also play a role in mediating the effects of vitamin D3 in cells [37,55]. Therefore, we decided to obtain further insights into the role of RORα and RORγ receptors in fibroblast activities. Human fibroblasts treated with 20(OH)3, 20,23(OH)_2_D3, 1,20(OH)_2_D3 or 1,20,23(OH)_3_D3 showed reduced proliferation in comparison to vehicle-treated cells. This effect was comparable to that seen for the positive control following treatment with 1,25(OH)_2_D3. The knockdown of either RORα or RORγ removed the inhibitory effects of vitamin D3 hydroxymetabolites. Furthermore, the inhibitory effect of secosteroids on collagen expression was not only abrogated, but the expression of some collagen types in the si-RORs cells, especially *COL3A1* in si-RORγ cells, increased. Since the knockdown of RORα or RORγ either removed the effect of secosteroids on the expression of the collagen genes in fibroblasts, or sometimes caused the opposite effect to that seen on control cells, we conclude that these receptors play a role in the action of vitamin D3 hydroxymetabolites on collagen synthesis by fibroblasts. Contrarily to *COL1A1, COL1A2* and *COL3A1*, the expression of *FN1, ACTA1, THBS1* and *PDGFA* were decreased in control fibroblasts upon treatment with the CYP11A1-derived vitamin D3 hydroxymetabolites, whereas it was enhanced (slightly or strongly) in fibroblasts lacking RORα or RORγ. Metallopeptidase (MMP1) has a role in the degradation of extracellular matrix proteins [80]. Treatment with vitamin D3 hydroxymetabolites stimulated the expression of *MMP1* in control cells, whereas *MMP1* expression was reduced in si-RORα and si-RORγ cells by the same compounds. The expression of the anti-inflammatory genes under study, including *IL-6*, *IL-8, IL-33* and *TGFB1*, decreased in control fibroblasts after treatment with vitamin D hydroxymetabolites. The knockdown of either RORα or RORγ led to pro-inflammatory effects of vitamin D3 hydroxymetabolites. These results indicate not only the involvement of RORα and RORγ in pro-fibrotic and pro-inflammatory functions in dermal fibroblasts, but also identify them as targets for novel secosteroids, which, through reverse agonism (the inhibition of the transcriptional activity of RORs after binding to the receptors), can inhibit fibrotic and inflammatory activities.

CYP27B1 can hydroxylate CYP11A1-derived vitamin D3 hydroxymetabolites at C1α (Figure 4A), modifying their biological activity through an increased affinity for the VDR [18,55]. For example, 20(OH)D3 lacks calcemic activity whereas 1,20(OH)_2_D3 does display moderate calcemic activity, although this is lower than for 1,25(OH)_2_D3 [25]. Therefore, whether 20(OH)D3 and 20,23(OH)_2_D3 act directly on the VDR (or RORs) to cause their biological effects or whether they must first undergo 1α-hydroxylation is an important question. To explore whether the antifibrotic activities of 20(OH)D3 and 20,23(OH)_2_D3 derivatives are dependent on hydroxylation at the C1α position, we silenced the *CYP27B1* gene in human fibroblasts. Proliferation, as well as collagen synthesis and the expression of *COL1A1* and COL3A1 genes, were inhibited in the control cells (WT) by these two secosteroids. The two secosteroids also retained the ability to downregulate the expression of *TGFB1* and *THBS1* genes in si-CYP27B1 fibroblasts. Thus, we conclude that the C1α-hydroxylation of 20(OH)D3 and 20,23(OH)_2_D2 by CYP27B1 to produce1,20(OH)_2_D3 and 1,20,23(OH)_3_D3, respectively, is not required for their actions on fibroblasts. Similar results have been reported for 20(OH)D2 in keratinocytes, where the silencing of the *CYP27B1* gene did not prevent this secosteroid from the stimulation of the keratinocyte differentiation program [23]. 

In this report, we provide an insight into the mechanism of action the vitamin D hydroxyderivatives with the involvement of VDR, RORα and RORγ. The latter conclusion is strengthened by the previous demonstration of the requirement for RORγ in RORC^−/−^ mice [55]. We also acknowledge the limitations of our studies. Recent molecular modeling and functional studies have identified additional nuclear receptors for CYP11A1-derived vitamin D3 hydroxyderivatives, such as the aryl hydrocarbon receptor (AhR) [59,81,82], liver X receptors (LXRα and β) [59,83] and peroxisome proliferator-activated receptor gamma receptor (PPAR-γ) for tachysterol [40]. Therefore, the challenge for future research is determining the degree to which these additional receptors are involved in the regulation of fibroblast activities. In our opinion, such effects would be dependent on the concentration of the ligands, their metabolic transformation, interactions between different nuclear receptors and coupling to the downstream signal transduction pathways, with similar downstream signaling being responsible for the anti-inflammatory activities [84]. For example, CYP11A1-derived vitamin D3 hydroxyderivatives can inhibit nuclear factor kappa B NFκB [18] or IL-17 [37,75] activities or stimulate NRF-2 or P53 signaling [21]. Dissecting these activities and inter-receptor communication represent challenging but exciting new areas for future research.

## 5. Conclusions

The current study validates that the novel CYP11A1-derived vitamin D3 hydroxymetabolites have anti-fibrotic activities in human fibroblasts, supporting previous reports for murine fibroblasts in vitro and in vivo [17,55]. These effects are similar to those of 1,25(OH)_2_D3 and hydroxyl-pregnacalciferols (derivatives with a shortened side chain) [54]. Thus, non-calcemic vitamin D hydroxyderivatives are excellent candidates for treatment of local or systemic fibrosing diseases. In regard to the underlying mechanisms of the action of CYP11A1-derived vitamin D3 hydroxyderivatives on human fibroblasts in reducing fibrosis and inflammation, the present study demonstrates that such activities in human fibroblasts depend on the intact VDR, as well as on RORα and RORγ. In addition, although CYP27B1 might be important for the metabolism of these and other vitamin D3 derivatives in some settings, its presence is not required for the observed biological actions of 20(OH)D3 and 20,23(OH)_2_D3 on fibroblasts. This is an important finding, since in the classical vitamin D activation pathway, the precursor molecule 25(OH)D3 requires hydroxylation by CYP27B1 to generate biologically active 1,25(OH)_2_D3 [3,85]. The possibility that novel vitamin D3 hydroxymetabolites can also exert antifibrotic activities through action on additional nuclear receptors represents a future challenge. Such activity would be context-dependent and dependent on the intracellular ligand concentration. In summary, this study identifies VDR or RORα/γ receptors as promising targets for antifibrogenic and anti-inflammatory activities by CYP11A1-derived hydroxyderivatives in human fibroblasts, with CYP27B1 not being required for these effects. 

## Figures and Tables

**Figure 1 cells-13-00239-f001:**
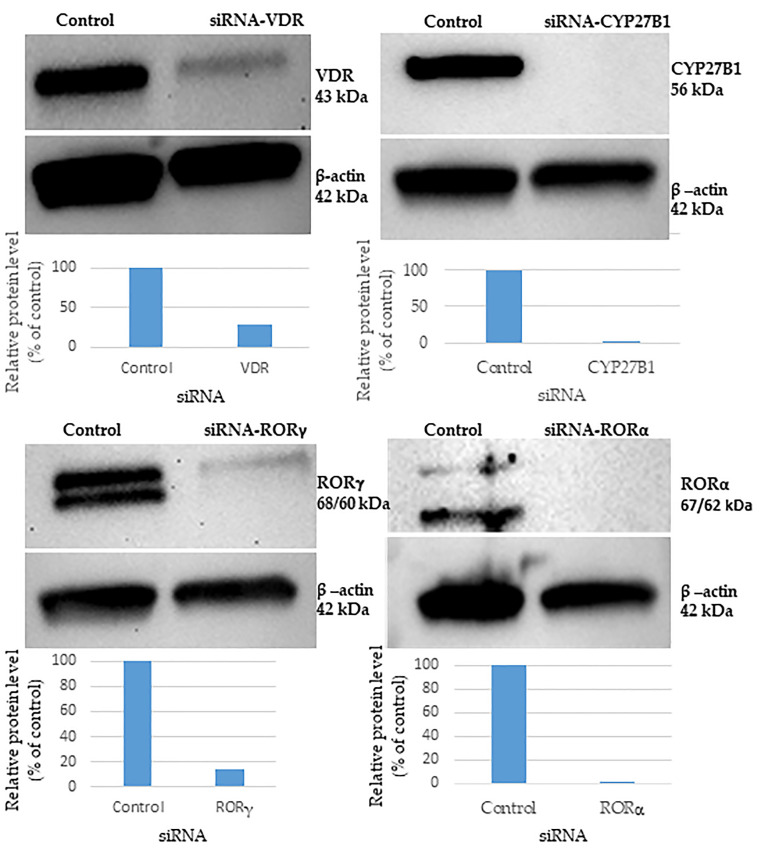
Knockdown efficiency evaluated by Western blotting. Human dermal fibroblasts were transfected with the appropriate siRNA, as indicated in the figure, and were incubated for 96 h. Cells were lysed and proteins isolated. Protein levels of VDR, CYP27B1, RORγ and RORα were analyzed using Western blotting. Beta-actin served as an internal control. Blots were analyzed using ImageJ and the intensity of each band was measured and normalized relative to beta-actin; this change is expressed as % of control, as shown in the histograms below each blot.

**Figure 2 cells-13-00239-f002:**
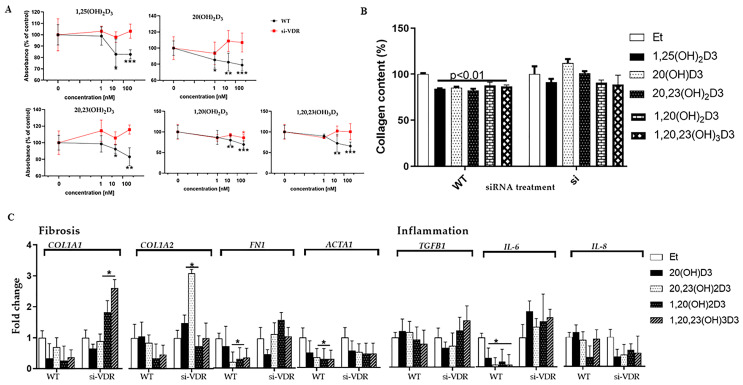
The VDR is required for the effects of vitamin D3 hydroxyderivatives on proliferation, collagen synthesis, fibrosis and inflammation. Control (WT) human dermal fibroblasts were compared to si-VDR cells. (**A**) Wild type or si-VDR cells were incubated with the secosteroids (100 nM) as indicated for 48 h and proliferation was measured using the MTS assay. Data are expressed relative to control cells treated with the ethanol vehicle (% of control) and are means ± SD (*n* ≥ 6), * *p* < 0.05, ** *p* < 0.01, *** *p* < 0.001, by *t*-test. (**B**) The supernatant was collected from WT or si-VDR cells treated with the secosteroids (100 nM) as indicated, for 24 h. Ethanol vehicle served as a negative control and 1,25(OH)_2_D3 as a positive control. Total collagen content of samples was determined using the Sircol collagen assay. Data were analyzed using the *t*-test. (**C**) A graph was constructed to show the expression of the genes involved in fibrosis and inflammation in WT and si-VDR fibroblasts. HDFn cells (WT or si-VDR) were treated with 100 nM secosteroids, as shown, for 24 h. RNA was isolated and the expression of genes involved in fibrosis or inflammation was measured using RT-PCR. Data are presented graphically as fold-changes relative to the ethanol vehicle for each cell type.

**Figure 3 cells-13-00239-f003:**
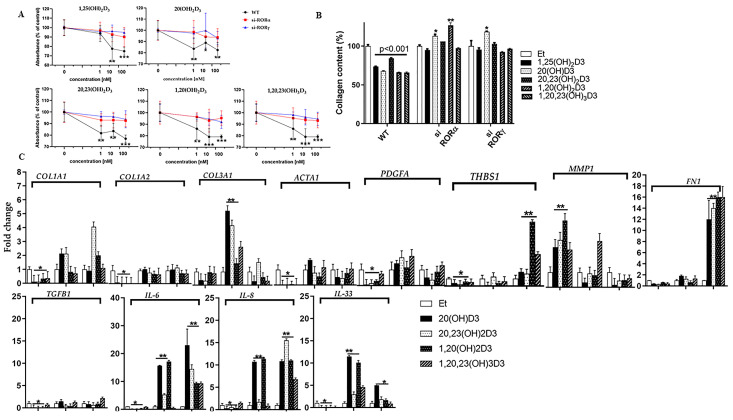
RORα/γ receptors are required for the effects of vitamin D3 hydroxyderivatives on proliferation, collagen synthesis, fibrosis and inflammation. (**A**) WT, si-RORα or si-RORγ cells were incubated with the secosteroids (1, 10 or 100 nM) or ethanol vehicle, as indicated, for 24 h, and proliferation was measured using the MTS assay. Data are expressed relative to control cells treated with the ethanol vehicle (% of control) and are means ± SD (*n* ≥ 6), * *p* < 0.05, ** *p* < 0.01, *** *p* < 0.001, by the *t*-test. (**B**) The supernatant was collected from WT or si- cells treated with the secosteroids (100 nM) as indicated, for 24 h. Ethanol vehicle served as a negative control and 1,25(OH)_2_D3 as a positive control. Total collagen content of samples was determined using the Sircol collagen assay. Data were analyzed using the *t*-test. * *p* < 0.05, ** *p* < 0.01. (**C**) Graphs were constructed to show the expression of the genes involved in fibrosis and inflammation in WT and si-ROR fibroblasts. Human dermal fibroblasts (WT, si-RORα or si-RORγ) were treated with 100 nM secosteroids, as shown, for 24 h. RNA was isolated and the expression of genes involved in fibrosis and inflammation was measured using RT-PCR. Data are presented as graphs showing fold-changes relative to the ethanol vehicle for each cell type.

**Figure 4 cells-13-00239-f004:**
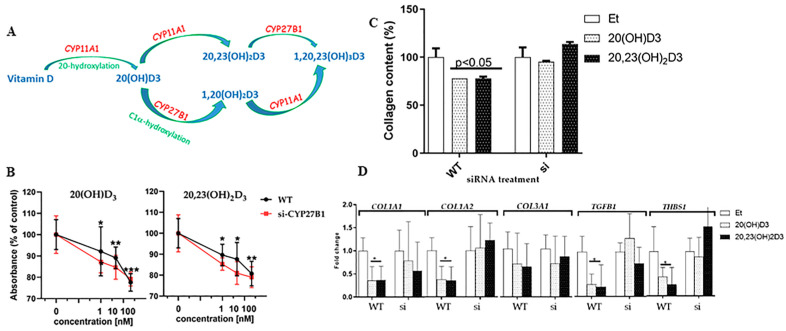
Expression of CYP27B1 is not required for the effects of 20(OH)D3 and 20,23(OH)_2_D3 on fibroblasts. (**A**) Pathways for the metabolism of vitamin D3 by CYP11A1 and CYP27B1. Enzymes are in red, derivatives are in blue and reactions are in green color. (**B**) WT (control) or si-CYP27B1 cells were incubated with 20(OH)D3 or 20,23(OH)_2_D3 (1, 10 or 100 nM) for 24 h, and proliferation was measured using the MTS assay. Data are expressed relative to control cells treated with the ethanol vehicle (% of control) and are means ± SD (*n* ≥ 6), * *p* < 0.05, ** *p* < 0.01, *** *p* < 0.001, by the *t*-test. (**C**) the supernatant was collected from WT or si-CYP27B1 cells treated with the secosteroids (100 nM) or the ethanol vehicle, as indicated, for 24 h. Total collagen content of samples was determined using the Sircol collagen assay. The data were analyzed using the *t*-test, with a *p* value of 0.05 or less indicating statistical significance. (**D**) Graphs were constructed to show the expression of the genes involved in fibrosis and inflammation in WT and si-CYP27B1 fibroblasts. Human dermal fibroblasts (WT or si-CYP27B1) were treated with 100 nM secosteroids, as shown, for 24 h. RNA was isolated and the expression of genes involved in fibrosis and inflammation was measured using RT-PCR. Data are presented as graphs showing fold-changes relative to the ethanol vehicle for each cell type.

**Table 1 cells-13-00239-t001:** Primers and sequences used for RT-PCR.

Gene	Description	Sequence	Reference
Cyclophilin B		TGTGGTGTTTGGCAAAGTTC GTTTATCCCGGCTGTCTGTC	
CYP27B1	cytochrome P450 family 27 subfamily B member 1	CTTGCGGACTGCTCACTG CGCAGACTACGTTGTTCAGG	
COL1A1	collagen I, type alpha 1	CAGGTCTCGGTCATGGTACCT TCGAGGGCCAAGACGAA	[60]
COL1A2	collagen I, type alpha 2	GCCCCCCAGGCAGAGA CCAACTCCTTTTCCATCATACTGA	[60]
COL3A1	collagen III, type alpha 1	CACTGGGGAATGGAGCAAAAC ATCAGGACCACCAATGTCATAGG	[61]
FN1	fibronectin	GGAGAATTCAAGTGTGACCCT CA TGCCACTGTTCTCCTACGTGG	[62]
GAPDH	glyceraldehyde 3-phosphate dehydrogenase	AGCCACATCGCTCAGACAC GCCCAATACGACCAAATCCC	
IL-6	cytokine of the IL-1 family	GAAGCTCTATCTCGCCTCCA AGCAGGCAACACCAGGAG	
IL-8	cytokine of the IL-1 family	AGACAGCAGAGCACACAAGC ATGGTTCCTTCCGGTGGT	
IL-33	cytokine of the IL-1 family	CACCCCTCAAATGAATCAGG GAGCTCCACAGAGTGTTCC	[63]
MMP1	metallopeptidase 1	CTGGGAAGCCATCACTTACCTTGC GTTTCTAGAGTCGCTGGGAAGCTG	[64]
PDGFA	platelet-derived growth factor, bone repair and regeneration	GCAGTCAGATCCACAGCATC TCCAAAGAATCCTCACTCCCTA	
RORA	RAR-related orphan receptor A	GTCAGCAGCTTCTACCTGGAC GTGTTGTTCTGAGAGTGAAAGGCACG	[37]
RORC	RAR-related orphan receptor C	CAGCGCTCCAACATCTTCT CCACATCTCCCACATGGACT	[37]
ACTA1 (α-SMA)	smooth muscle actin	CCGACCGAATGCAGAAG GA ACAGAGTATTTGCGCTCCGAA	[62]
VDR	vitamin D receptor	CTTACCTGCCCCCTGCTC AGGGTCAGGCAGGGAAGT	[23]
TGFB1	transforming growth factor beta 1, cytokines family	GCAGCACGTGGAGCTGTA CAGCCGGTTGCTGAGGTA	
THBS1	thrombospondin 1	CTG ATC TGG GTT GTG GTT GTA CCT GTG ATG ATG ACG ATG A	[65]

## Data Availability

Data are contained within the article.

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
