# Peer review of "Novel Vitamin D3 Hydroxymetabolites Require Involvement of the Vitamin D Receptor or Retinoic Acid-Related Orphan Receptors for Their Antifibrogenic Activities in Human Fibroblasts"

_cells, 2024, doi:10.3390/cells13030239_

Round 1

Reviewer 1 Report

Comments and Suggestions for Authors

Janjetovic et al. explores the involvement of vit D3 receptors (VDR) and retinoic acid-related orphan receptors (RORs) (RORa and RORg) in mediating the anti-proliferative and antifibrogenic role of CYP11A1-derived vit D3 using human neonatal dermal fibroblasts. Though the findings are novel, the experiments conducted and figure representations to prove the hypothesis are inconclusive. The manuscript is poorly written and contains several typological errors and redundant text. There are several points that need to be addressed. Some of them are listed below:

1.       The authors have established VDR & RORs are essential in mediating vitD3 activity. The silencing of CYP27B1, VDR and RORs has been used as an essential tool to study the antiproliferative and antifibrogenic activities, why do the authors exclude CYP11A1 from this study, which is the key enzyme responsible for hydroxylation of secosteroids ?

2.       The authors have used qRTPCR and biochemical assays to determine the expression of collagen. Ideally a western blot should be done to determine the protein expression either from the culture media supernatant or from whole cell lysates to confirm the collagen accumulation.

3.       Nowhere in the manuscript or supplemental data, the authors have included the raw data either for gene expression i.e., transcript level / relative transcript level, or amount of the collagen. Hence it is hard to review this article based on fold changes and percentages.

4.       It is surprising to see that the authors did not observe any intrinsic differences in VDR/RORa/RORc/CYP27B1 silenced fibroblasts. This has not been discussed.

5.       Do the authors have any rescue experiments to show the reversal observed effects?

6.       Authors should mention in the text, within how many hours / passages after gene silencing the experiments were performed.

7.       What was the basis for picking certain genes as representative markers for fibrosis or inflammation?  Why certain genes were dropped from analysis in VDR and CYP27B1 silenced studies?

8.       I lack certainty regarding the method used to create the heatmaps. Were the gene expressions conducted with biological replicates? In an ideal scenario, a fold change of '1' indicates equal expression. Consequently, I am puzzled by the presence of both red and white color codes for the value '1.' I am uncertain about the range indicator legend; what does the white color signify? Even if this serves as a method to depict the data, there is ambiguity.

9.       In scientific literature KO stands for ‘knockout’. I believe, the authors have done gene silencing ie knockdown experiments. So, this needs to be corrected to avoid any confusion for the readers.

10.   The western blot images of cyp27 b1 and RORa provided as a supplementary file dose not match the main figures.

11.   There are several typo errors I can list some of them, but the authors need to thoroughly check for such errors and redundant texts throughout the manuscript. For example:

a.       table 1 COL1A3 should be COL1A2

b.       subtitle 2.3 check the text

c.       subtitle 3.2, line number 15, needs to be rephrased for clarity.

d.       figure1 legend: please describe the method of western in method section only and not in figure legends.

e.       Please try to provide direct references for the convenience of readers in cross-referencing.

12.   Figures: overall some western blots lack quality. The densitometric plots are poorly represented and need improvement. Please provide the axis legends. Add molecular wt. indications on the WB.

13.   Provide the source of procurement of the hydroxy derivatives in the method section. Page 3, line 3.

Comments on the Quality of English Language

The manuscripts should be thoroughly checked for redundant texts and several sentences needs to be rephrased for clarity.

Author Response

We thank the reviewer for the effort to improve the manuscript and greatly appreciated the detailed critique.  The manuscript was revised as requested, some figures corrected, supplemental file changed. Changes in the manuscript are indicated by tracking tools or red font. Below is point by point response to the critique. Corresponding changes are in the manuscript.

Janjetovic et al. explores the involvement of vit D3 receptors (VDR) and retinoic acid-related orphan receptors (RORs) (RORa and RORg) in mediating the anti-proliferative and antifibrogenic role of CYP11A1-derived vit D3 using human neonatal dermal fibroblasts. Though the findings are novel, the experiments conducted and figure representations to prove the hypothesis are inconclusive. The manuscript is poorly written and contains several typological errors and redundant text. There are several points that need to be addressed. Some of them are listed below:

  1. The authors have established VDR & RORs are essential in mediating vitD3 activity. The silencing of CYP27B1, VDR and RORs has been used as an essential tool to study the antiproliferative and antifibrogenic activities, why do the authors exclude CYP11A1 from this study, which is the key enzyme responsible for hydroxylation of secosteroids ?

This is because we are investigating exogenous secosteroids, some of which are known to be in human serum and may also come from surrounding cells.  We are not examining the effects of endogenously produced secosteroids. Knocking down CYP11A1 will not stop the cells from producing 1,25(OH)2D3 which has some similar effects to the CYP11A1-derived secosteroids making such experiments difficult if not impossible to interpret. In addition, knocking CYP11A1 will inhibit endogenous gluocorticosteroidogenesis in fibroblasts generating pleiotropic effects.

  1. The authors have used qRTPCR and biochemical assays to determine the expression of collagen. Ideally a western blot should be done to determine the protein expression either from the culture media supernatant or from whole cell lysates to confirm the collagen accumulation.

Collagen was measured in cell supernatants, not by western blots but by a specific biochemical assay used by the majority of experts in the field.  Western blots of supernatant to measure collagen are not used since they produce a high background and are less precise as the majority of researchers in this field agree.

Nowhere in the manuscript or supplemental data, the authors have included the raw data either for gene expression i.e., transcript level / relative transcript level, or amount of the collagen. Hence it is hard to review this article based on fold changes and percentages.

We have changed the PCR data presentation using traditional graphs instead of heat maps.

  1. It is surprising to see that the authors did not observe any intrinsic differences in VDR/RORa/RORc/CYP27B1 silenced fibroblasts. This has not been discussed.

Per request, we have used the raw data from several different experiments. We compared ethanol treated cells (no-siRNA vs siRNA treated). MTS reagent was used to measure proliferation activity. It was added to the cells after two days of incubation with ethanol (solvent) (n=6).The absorbance at 490 nm was recorded. The results are presented in the form of table for a comparison. Since, we do not observe statistically significant differences in cell proliferation in presence or absence of the specified genes, we did not include these data in the manuscript.

                           VDR

                           CYP27B1

                          RORa

       RORg

No siRNA

siRNA

No siRNA

siRNA

No siRNA

siRNA

No siRNA

1.70±0.15

1.90±0.48

2.26±0.61

1.79±0.5

2.13±0.13

1.86±0.22

1.04±0.47

0.82±0.22

0.73±0.29

0.61±0.16

0.69±0.27

0.71±0.47

3.09±0.06

2.02±0.96

3.60±0.29

2.97±0.41

1.89±0.41

3.32±0.41

0.71±0.11

1.69±0.12

1.27±0.16

  1. Do the authors have any rescue experiments to show the reversal observed effects?

We did not perform rescue experiments for VDR and RORs as we were focused on the already demanding experiments with KO cells vs WT, believing that rescue experiments will provide only incremental information. As relates to CYP27B1 knockdown, there was no effect to reverse this.

  1. Authors should mention in the text, within how many hours / passages after gene silencing the experiments were performed.

Under Material and Methods, section 2.1 the following sentence is added:

Both western blotting and RT-qPCR were used to evaluate the efficiency of the knockdown of gene expression in cells passaged three times post- siRNA transfection.

  1. What was the basis for picking certain genes as representative markers for fibrosis or inflammation?  Why certain genes were dropped from analysis in VDR and CYP27B1 silenced studies?

We have selected genes with well established roles in fibrosis or inflammatory processes. We also have included the genes that are expressed consistently in different experiments.

  1. I lack certainty regarding the method used to create the heatmaps. Were the gene expressions conducted with biological replicates? In an ideal scenario, a fold change of '1' indicates equal expression. Consequently, I am puzzled by the presence of both red and white color codes for the value '1.' I am uncertain about the range indicator legend; what does the white color signify? Even if this serves as a method to depict the data, there is ambiguity.

We have changed the PCR data presentation using traditional histogram graphs instead of heat maps to avoid confusion.

  1. In scientific literature KO stands for ‘knockout’. I believe, the authors have done gene silencing ie knockdown experiments. So, this needs to be corrected to avoid any confusion for the readers.

We have replaced “KO” with “si” (silenced) for a clearer understanding.

  1. The western blot images of cyp27 b1 and RORa provided as a supplementary file dose not match the main figures.

We carefully checked original blots and labeled them appropriately. We included these blots in its full size just for the reviewers. The selected parts of the blots we presented and described in Figure 1.

  1. There are several typo errors I can list some of them, but the authors need to thoroughly check for such errors and redundant texts throughout the manuscript. For example:

We have removed several sentences of redundant text from the Results where experimental conditions were described in both the Results text and Figure legends

  1. table 1 COL1A3 should be COL1A2

Thank you for noticing this typo. It is fixed.

  1. subtitle 2.3 check the text

Thank you for noticing this typo. It is fixed.

  1. subtitle 3.2, line number 15, needs to be rephrased for clarity.

Thank you for noticing this typo. It is fixed.

  1. figure1 legend: please describe the method of western in method section only and not in figure legends.

Figure 1 legend is updated:

Figure 1. Knock-down efficiency evaluated by western blotting. Human dermal fibroblasts were transfected with the appropriate siRNA as indicated in the figure, and incubated for 96 h. Cells were lysed and proteins isolated. Protein levels of VDR, CYP27B1, RORg and RORa were analyzed using western blotting. Beta actin served as an internal control. Blots were analyzed using Image J and the intensity of each band was measured and normalized relative to the beta-actin; this change is expressed as % of control, as shown in the histograms below each blot.

  1. Please try to provide direct references for the convenience of readers in cross-referencing.

We apologize but do not understand the critique. The proper references are cited where applicable.

  1. Figures: overall some western blots lack quality. The densitometric plots are poorly represented and need improvement. Please provide the axis legends. Add molecular wt. indications on the WB.

Figure 1 is updated. Please see the improvement with added corresponding molecular masses and the description of densitometry plots with relabeled axes.

  1. Provide the source of procurement of the hydroxy derivatives in the method section. Page 3, line 3.

This was actually provided.The hydroxyderivatives were synthesized in Dr Wei Li and Dr Tuckey’s labotatories and the description is provided within the references to the synthesis in the Materials and Methods (Paragraph 1).

Comments on the Quality of English Language

The manuscripts should be thoroughly checked for redundant texts and several sentences needs to be rephrased for clarity.

Thank you for the critique. The manuscript was thoroughly checked by a native English speaker who is senior co-author, Dr Tuckey.

Reviewer 2 Report

Comments and Suggestions for Authors

In this manuscript, the examined the effects of CYP11A1-derived vitamin D metabolites (20(OH)D3, 20,23(OH)2D3, 1,20(OH)2D3 and 1,20,23(OH)2D3) on proliferation, collagen production and expression of fibrotic and inflammatory genes in human fibroblasts where VDR, RORa, RORg or CYP17B1 was knocked down. These vitamin Ds inhibited proliferation, collagen production and expression of fibrotic and inflammatory genes. Interestingly, these effects were abolished in cells with VDR, RORa, or RORg knockdown, where CYP27B1 knockdown did not change the effects of vitamin D metabolites. Although the findings are interesting, I have the following questions and comments.

1. Fig.1. The effect of knockdown of VDR on VDR expression was shown. How about that on expression of RORa, RORg and CYP27B1? I have similar questions about RORa, RORg and CYP27B1 knockdown.

2. Did knockdown of VDR, RORa, RORg and CYP27B1 have any effect on proliferation of cells in the absence of vitamin D treatment?

3. Vitamin D metabolites decreased expression of fibrotic and inflammatory gene expression in WT cells, but increased expression of some of these genes in cells with VDR, RORa or RORg knockdown. Please discuss what factors mediate the profibrotic/proinflammatory effect of vitamin D metabolites in these cells in more detail, although the authors mention AHR, LXRs and PPARg in Conclusion.

4. "5. Conclusion" is too long. Discussion and speculation should be moved to "4. Discussion".

Author Response

We thank the reviewer for the effort to improve the manuscript and greatly appreciated the detailed critique.  The manuscript was revised as requested, some figures corrected, supplemental file changed. Changes in the manuscript are indicated by tracking tools or red font. Below is point by point response to the critique. Corresponding changes are in the manuscript.

In this manuscript, the examined the effects of CYP11A1-derived vitamin D metabolites (20(OH)D3, 20,23(OH)2D3, 1,20(OH)2D3 and 1,20,23(OH)2D3) on proliferation, collagen production and expression of fibrotic and inflammatory genes in human fibroblasts where VDR, RORa, RORg or CYP17B1 was knocked down. These vitamin Ds inhibited proliferation, collagen production and expression of fibrotic and inflammatory genes. Interestingly, these effects were abolished in cells with VDR, RORa, or RORg knockdown, where CYP27B1 knockdown did not change the effects of vitamin D metabolites. Although the findings are interesting, I have the following questions and comments.

  1. Fig.1. The effect of knockdown of VDR on VDR expression was shown. How about that on expression of RORa, RORg and CYP27B1? I have similar questions about RORa, RORg and CYP27B1 knockdown.

The effect of VDR knockdown on VDR expression is shown in Figure 2. Similarly the effects of RORa, RORg knockdown are shown in Figure 3 and CYP27B1 in Figure 4.

  1. Did knockdown of VDR, RORa, RORg and CYP27B1 have any effect on proliferation of cells in the absence of vitamin D treatment?

Per request, we have used the raw data from several different experiments. We compared ethanol treated cells (no-siRNA vs siRNA treated). MTS reagent was used to measure proliferation activity. It was added to the cells after two days of incubation with ethanol (solvent) (n=6).The absorbance at 490 nm was recorded. The results are presented in the form of table for a comparison. Since, we do not observe statistically significant differences in cell proliferation in presence or absence of the specified genes; we did not include these data in the manuscript.

                           VDR

                           CYP27B1

                          RORa

       RORg

No siRNA

siRNA

No siRNA

siRNA

No siRNA

siRNA

No siRNA

1.70±0.15

1.90±0.48

2.26±0.61

1.79±0.5

2.13±0.13

1.86±0.22

1.04±0.47

0.82±0.22

0.73±0.29

0.61±0.16

0.69±0.27

0.71±0.47

3.09±0.06

2.02±0.96

3.60±0.29

2.97±0.41

1.89±0.41

3.32±0.41

0.71±0.11

1.69±0.12

1.27±0.16

  1. Vitamin D metabolites decreased expression of fibrotic and inflammatory gene expression in WT cells, but increased expression of some of these genes in cells with VDR, RORa or RORg knockdown. Please discuss what factors mediate the profibrotic/proinflammatory effect of vitamin D metabolites in these cells in more detail, although the authors mention AHR, LXRs and PPARg in Conclusion.

The discussion has been added and  results have been explained in a more detailed fashion.

  1. "5. Conclusion" is too long. Discussion and speculation should be moved to "4. Discussion".

Thank you for critique. The discussion and speculation sections were modified and moved  to the discussion from the Conclusions. This has improved both Discussion and Conclusion sections. Thank you for your critique.

The manuscripts should be thoroughly checked for redundant texts and several sentences needs to be rephrased for clarity.

Thank you for the critique. The manuscript was thoroughly checked by a native English speaker who is senior co-author, Dr Tuckey.

Round 2

Reviewer 1 Report

Comments and Suggestions for Authors

The manuscript still requires minor corrections for typos and data presentation, example:

1. Fig 2 legend: KO should be corrected.

2. Subtitle3.3  needs clarity. 

3. Figure resolutions should be improved for readability.

4. In the previous version, siRORg cells treated with 20(OH)D3 had a 62.4 fold high expression of FN1 and 33.2 fold high expression of IL-6. In the new revised figure the numbers have changed. Why ?

My previous comment regarding this has not been addressed. I would again suggest the authors to provide raw values as supplementary tables so that readers can have clarity.

Comments on the Quality of English Language

Minor text and figure edit required

Author Response

We greatly appreciate the effort of the re viewer to improve our manuscript. It has been revised following the reviewer's critique. Below is point by point response to the critique.

The manuscript still requires minor corrections for typos and data presentation, example:

  1. Fig 2 legend: KO should be corrected.

Thank you for noticing this error. It is corrected.

  1. 3  needs clarity. 

The subtitle has been revised as follows:

3.3. RORa and RORg  are required for phenotypic effects of vitamin D hydroxyderivatives on fibroblasts

  1. Figure resolutions should be improved for readability.

We have changed the resolution and improved the quality.

  1. In the previous version, siRORg cells treated with 20(OH)D3 had a 62.4 fold high expression of FN1 and 33.2 fold high expression of IL-6. In the new revised figure the numbers have changed. Why ?

My previous comment regarding this has not been addressed. I would again suggest the authors to provide raw values as supplementary tables so that readers can have clarity.

We thank the reviewer for detailed analysis and critique.  We have made an error in previous version of the manuscript, which we corrected.

For the clarity, we enclose raw PCR data in the form of a table, see attachment.

Comments on the Quality of English Language

Minor text and figure edit required

Thank you for your comments. The text and figures were revised, where applicable. In addition, the manuscript was proofread by a native English speaker.

Reviewer 2 Report

Comments and Suggestions for Authors

All comments have been addressed.

Author Response

We thank the reviewer for his final approval and time and effort required to improve our presentation.